# The combination of active partitioning and toxin-antitoxin systems is most advantageous for low-copy plasmid fitness

Johannes Effe[1], Mario Santer[1], Yiqing Wang[1,2], Theresa E. Feenstra[1], Nils F. Hülter [1] & Tal Dagan [1] ✉

Extrachromosomal and mobile genetic elements, including plasmids, and accessory chromosomes, are prevalent in all life domains. Elements integrated into the host chromosome replicate and segregate via the host life cycle. In contrast, the persistence of autonomously replicating elements relies on their ability to remain within the host population. Here, we compare the evolutionary advantage of different persistence strategies found in prokaryotic plasmids. Through intracellular competitions between plasmid genotypes, we find that the combination of active partitioning during cell division with a toxin-antitoxin (TA) system for post-segregational killing increases plasmid fitness more than either strategy alone. Mathematical modeling of long-term plasmid evolution, calibrated with empirical plasmid loss dynamics, further supports these findings. A survey of enterobacterial genomes indicates that partitioning and TA systems are core features of large plasmids. Indeed, we confirm the presence of a previously unrecognized type I TA system in conjugative IncX3 plasmids, which serve as important vectors of antibiotic resistance in human pathogens. These findings suggest that large plasmids – including conjugative and mobilizable types – encode TA systems, some of which have yet to be identified. The combination of TA and partitioning systems emerges as the most effective strategy for the evolutionary success of low-copy extrachromosomal elements.

Plasmids are extrachromosomal genetic elements that reside in prokaryote cells. The evolutionary success of plasmids in bacterial populations following invasion depends on drift and selection that occur in two organizational levels: the hosting cell and the population of hosts[1–3]. An important determinant of plasmid fitness is the plasmid ability to replicate and segregate during cell division. Failure in plasmid replication into monomeric entities lowers the number of plasmid-copies available for segregation, which may lead to imbalanced segregation and plasmid loss[4,5]. Indeed, intracellular plasmid competitions showed that plasmid resolution into monomers is an important determinant of plasmid fitness[6]. Segregational plasmid loss leads to

the emergence of plasmid-free cells; hence, plasmid persistence furthermore depends on the plasmid effect on their host fitness[7–9]. Plasmids supplying the host with a beneficial trait, e.g., resistance to antibiotics, may confer their host a fitness advantage that promotes plasmid persistence. However, in the absence of selection pressure for the plasmid-encoded trait, plasmids that are neutral, or slightly deleterious, to their host fitness are at risk of extinction due to competition for resources with plasmid-free cells[10,11].

Plasmid persistence can be promoted by active partition systems that facilitate plasmid segregation via their effect on plasmid localization within the host cell. The systems typically comprise three main

[1]Institute of General Microbiology, Kiel University, Kiel, Germany. [2]Present address: Institut Pasteur, Université de Paris Cité, CNRS UMR3525, Microbial Evolutionary Genomics, Paris, France. ✉e-mail: tdagan@ifam.uni-kiel.de

components: a partition binding site (centromere), a centromere binding protein (CBP) and a nucleotide triphosphatase (NTPase). The partition complex is formed when the CBP binds the plasmid centromere(s) and interacts with the NTPase, which drives plasmid transport and positioning[12]. The presence of partition systems is associated with the plasmid size. For example, in *Enterobacteria*, plasmids larger than 25 kbp typically harbor a partition system, which facilitates their transition via the nucleoid area and thus promotes their segregation[13]. Plasmids of that size typically have a low copy number, e.g., <5 copies in *Escherichia* plasmids[14]. Plasmid partition systems likely share a common origin with chromosomal partition systems that have a similar function, but are evolutionary diverged[15].

Another route for plasmid persistence is plasmid-encoded toxin-antitoxin (TA) systems that inhibit the growth of plasmid-free cells following segregational loss. Toxin-antitoxin systems typically comprise a toxin that inhibits an essential cellular process and an antitoxin that neutralizes its cognate toxin. TA systems are found also in bacterial chromosomes (or phages), where they assume diverse functions, including the prokaryotic immune system and survival in growth-limiting conditions[16,17]. The toxin is commonly a protein that induces a toxic effect on the cell, e.g., via competitive binding or degradation of essential proteins. The antitoxin can be either an RNA or a protein. In type I systems, an RNA antitoxin neutralizes a proteinaceous toxin by binding the toxin mRNA, thus inhibiting the toxin translation. Example is the plasmid-encoded *hok*/*sok* TA post-segregational killing[18]. Plasmids may harbor multiple TA systems and recent studies indicate a role of transposable elements in the evolution of plasmid TA repertoire[19,20]. Toxin-antitoxin and active partition systems increase plasmid stability via two different strategies[21] and their combination has been observed, e.g., in enteric virulence plasmids[22,23].

In this work, we ask which (if any) of these plasmid traits is superior over the other. For that purpose, we conducted intracellular plasmid competitions between initially unstable non-mobile plasmids harboring either system or both. Both systems in our experimental setup were adopted from plasmid R1 that was originally isolated from a clinical *Salmonella* isolate[24]. The *parA* locus in R1 was adopted as the partition system; it includes the CBP-encoding *parR*, the actin-like ATPase *parM* and the ParR binding site *parC*[25]. The *parB* locus of R1, which includes the *hok*/*sok* toxin-antitoxin system was adopted as the TA system[26,27]. The Hok (Host killing) protein is a toxin that binds the cell membrane, thereby inhibiting the cell respiration. The antitoxin sok (suppression of killing) is an sRNA that inhibits *hok* translation by binding to its mRNA[28]. Using the empirical results from plasmid competition, we developed a mathematical model for the effect of partition and TA systems on plasmid stability on evolutionary timescales. Finally, we validate our prediction by observing the distribution of partition and TA systems in contemporary plasmids.

## Results

### Intracellular head-to-head competitions enable the comparison of plasmid fitness

To compare the effect of TA and Par systems on plasmid stability, we established an experimental system for intracellular plasmid competitions. The recipient population is first transformed with the two competing plasmids simultaneously. The presence of the competing plasmids is confirmed by imposing selection for antibiotic resistance markers in the competing plasmids. The recipient population is then incubated for the competition duration with five replicates in the absence of antibiotics. At the end of the competition, the relative frequency of cell types in the replicate populations is determined (Fig. 1a). The plasmid that is found in a higher frequency of homoplasmic hosts is considered to have a higher fitness compared to the competing plasmid. This system enables us to rank plasmid traits pairwise according to their relative fitness.

As the low and high reference for plasmid stability, we utilized two pCON plasmid genotypes that are characterized by either low stability (unstable, U) or high stability (stable, S) (Fig. 1b). The backbone of both model plasmids originated from the pBBR1 broad-host range plasmid[29]; both plasmids are non-mobile. The pCON instability arises from conflicts between gene transcription and plasmid replication, which promotes the formation of plasmid multimers and result in segregational loss. The stable pCON genotype has a slightly greater distance between the origin of replication and the antibiotic resistance gene promoter, leading to a higher frequency of monomeric plasmids and high plasmid stability[30]. Competing plasmids harboring a toxin-antitoxin system (T), partition system (P), or both systems (PT) were constructed using the unstable pCON plasmid backbone. To validate the effect of the plasmid stability system, all model plasmids were constructed with alternative selection markers: either *nptII* that confers resistance to kanamycin, or *cat* that confers resistance to chloramphenicol (Fig. 1b), which we denote by pCONn and pCONc, respectively, followed by the stability trait, e.g., pCONn-P (Fig. 1b)[6]. The model plasmids are therefore identical, with the exception of the antibiotic resistance markers and the in/stability module. The model plasmids have a no significant effect on their host fitness with the exception of pCONn-T and pCONc-U that have a slightly deleterious effect on their host fitness (mean $w_{pCONn-T} = 0.94 \pm 0.015$ SE; mean $w_{pCONc-U} = 0.95 \pm 0.016$ SE; Supplementary Fig. S1). The model plasmid size ranges between 2733 bp (pCONc-S) and 5273 bp (pCONn-PT), and the plasmid copy number ranged between 2.4 and 8.8 plasmids (relative to the chromosome; Supplementary Table S1).

The stability of all model plasmids was evaluated from plasmid loss frequencies in a serial transfer experiment over five days. The unstable plasmid loss frequency was ca. 7–8% per transfer, on average, with little variation between pCONn-U and pCONc-U. The stable plasmids had an average loss frequency <1% for both plasmids (pCONn-S and pCONc-S) (Fig. 1c). The proportion of hosts after five days shows that the stability of all model plasmids equipped with a stability system was higher compared to the unstable model plasmids and similar to that of the stable model plasmid (Fig. 1c). Furthermore, the model plasmid stability was similar between plasmids harboring the same stability system (or none) that differ in their antibiotic resistance marker.

We first demonstrate our approach for pairwise plasmid fitness ranking with the results of plasmid competitions between the unstable and stable plasmid variants. The competition between pCONn-U and pCONc-S show that most replicate populations were dominated by homoplasmic hosts of the stable plasmid, with a smaller fraction of homoplasmic pCONn-U hosts ($P = 7.75 \times 10^{-7}$, using paired sign test, $n = 25$) (Fig. 1d). The proportion of plasmid-free cells (non-hosts) ranged between 0 and 29%. In several replicates, we observed heteroplasmic hosts, where both competing plasmids co-exist. Plasmid co-existence in our experimental system can be the results of either incomplete segregation of the competing plasmids or their fusion into heteromultimers[6]. To validate the effect of plasmid stability on the competition result, we also performed a reciprocal competition, pCONc-U and pCONn-S, using model plasmids where the antibiotic resistance markers are exchanged between the two plasmid backbones (Fig. 1b). The reciprocal competition yielded a majority of pCONn-S in most replicates, albeit the overall difference in the frequency of the competing plasmids was smaller ($P = 0.009$, using paired sign test, $n = 19$; Fig. 1d). Further competition between the two unstable plasmids pCONn-U and pCONc-U show no significant difference in their fitness ($P = 0.115$, using paired sign test; Fig. S2a), hence the negative fitness effect of pCONc-U on their host had a small effect on the plasmid competition outcome. Taken together, the competitions among the stable and unstable plasmids demonstrate a higher fitness of the stable plasmid, regardless of the antibiotic resistance selection marker in the plasmid backbone.

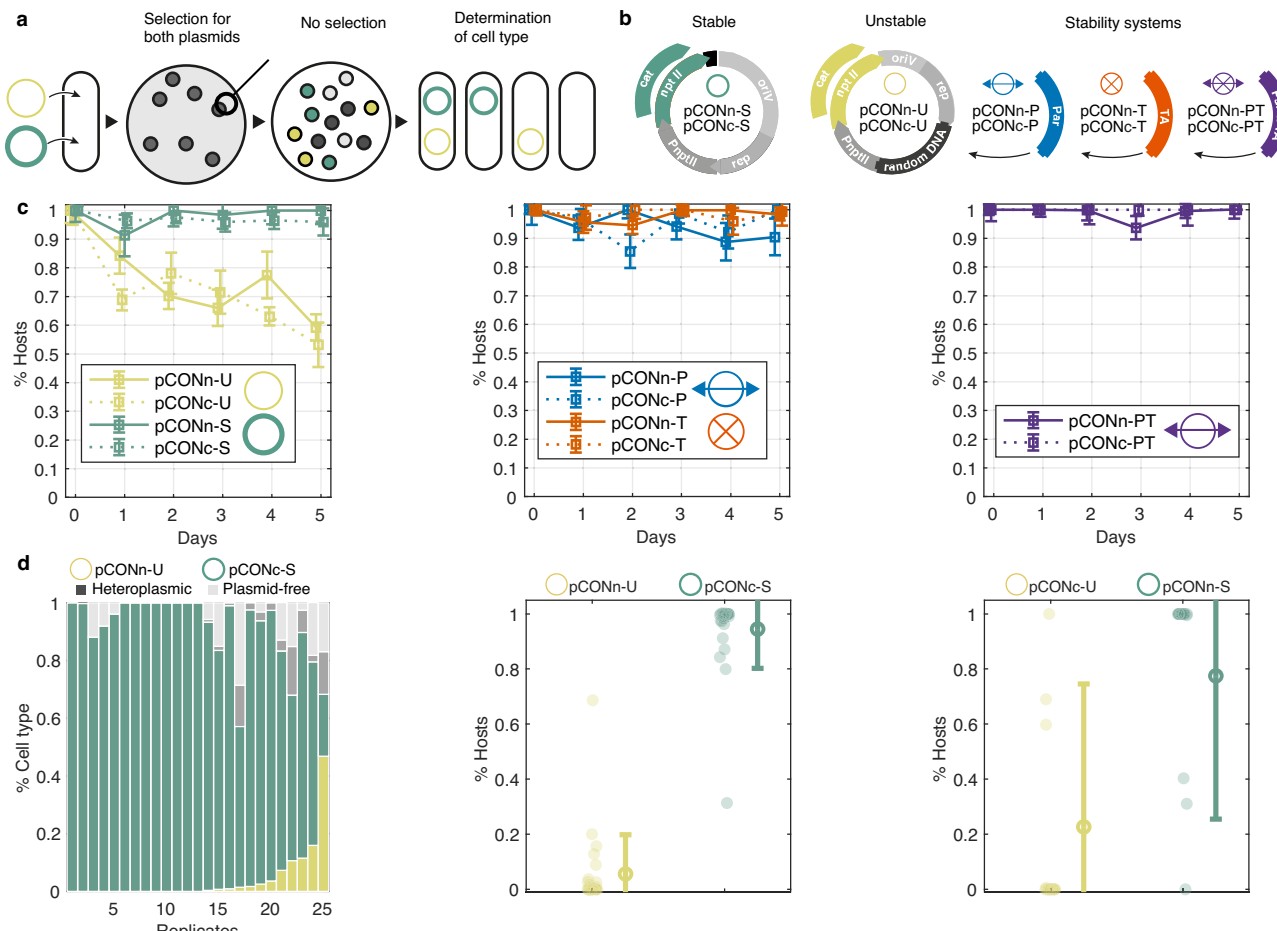

**Fig. 1 | An experimental system for pairwise in cellulo competitions between plasmid genotypes. a** The plasmids are introduced simultaneously into a naïve host population that is cultured under selection for both plasmid genotypes. Replicate colonies are picked and cultivated in non-selective conditions. The result of the plasmid competition is evaluated according to the frequency of host types at the end of the experiment. These include plasmid-specific hosts, heteroplasmic hosts of both plasmids and plasmid-free cells. **b** The set of competitor model plasmids. The plasmid backbone consists of the pBBR1 oriV and *rep* along with the P$_{nptII}$ promoter, followed by an *nptII* or *cat* antibiotic resistance gene. In the pCON-S variants, non-coding DNA between the oriV and the antibiotic resistance gene. The remaining plasmids vary in the locus between the *rep* gene and the pnptII promoter. In the pCON-U variants, a segment of random DNA is inserted; in the pCON-P plasmid, the parA locus of plasmid R1 is inserted, which contains the genes *parM*, *parR* and the parC DNA binding site; the plasmids pCONn-T and pCONc-T both contain the parB

locus of plasmid R1, which consists of the *hok/sok* toxin antitoxin system; plasmids of the PT type contain both the parA and the parB locus. Plasmid symbols are shown next to the plasmid types. The model plasmid size ranges between 2733 bp (pCONc-S) and 5273 bp (pCONn-PT) (see Supplementary Table S1 for details). **c** Model plasmid stability as evaluated in a serial transfer experiment under non-selective conditions ($n = 8$ replicates except for pCONc-R with $n = 5$ and pCONc-PT with $n = 7$ with error bars corresponding to standard deviation). **d** (left) A bar graph showing the results of pCONc-U vs pCONn-S competition. Each bar corresponds to a replicate population, with the shaded area in the bar proportional to the proportion of cell types in the population. (middle) The results of pCONc-U vs pCONn-S presented as a scatter plot. Host frequencies are shown with dots. The mean frequency is shown with a circle, with error bars corresponding to standard deviation ($n = 25$ replicates). (right) Results of the reciprocal competition between pCONn-U vs pCONc-S ($n = 25$ replicates). Source data are provided as a Source Data file.

## Pairwise competitions reveal superiority of plasmids harboring both partition and TA systems

To evaluate the effect of an active partition system on plasmid fitness, the plasmid pCONc-P was competed against the unstable pCONn-U and the stable pCONn-S. The competition against the unstable plasmid revealed that the proportion of homoplasmic pCONc-P hosts was nearly 100%, with only rare homoplasmic pCONn-U hosts across all replicates. In contrast to that, the competition against the stable plasmid yielded mostly homoplasmic pCONn-S hosts and only a minor proportion of homoplasmic pCONc-P hosts (Fig. 2a.1, 2). The reciprocal competitions between plasmids, where the stability systems are exchanged between the plasmid backbones, revealed no significant difference in the proportion of pCONn-P homoplasmic hosts in the competition against pCONc-U (statistical evaluation of the competition results is reported in Fig. 2b; Supplementary Fig. S3). The reciprocal competition of pCONn-P against the stable plasmid (pCONc-S)

yielded a majority of pCONc-S homoplasmic hosts (Supplementary Fig. S3), thus confirming the advantage of the stable plasmid over the plasmid comprising a partition system.

The competition of plasmids harboring a TA system against the unstable plasmids yielded a significantly higher frequency of pCONc-T homoplasmic hosts compared to pCONn-U hosts (Fig. 2a.3). The fitness advantage of the TA-harboring plasmid compared to the unstable plasmid was further confirmed in the reciprocal competition between pCONn-T and pCONc-U (Supplementary Fig. S3; both plasmids had comparable negative fitness effect on their host). These results are in agreement with previous results on the ability of plasmids harboring TA to outcompete a TA-lacking, otherwise identical plasmid[31,32]. In contrast, the competition between plasmid pCONc-T and the stable plasmid pCONn-S yielded similar proportions of homoplasmic hosts of the two plasmids, with pCONn-S having a slightly higher mean proportion of homoplasmic hosts (Fig. 2a.4). The reciprocal competition

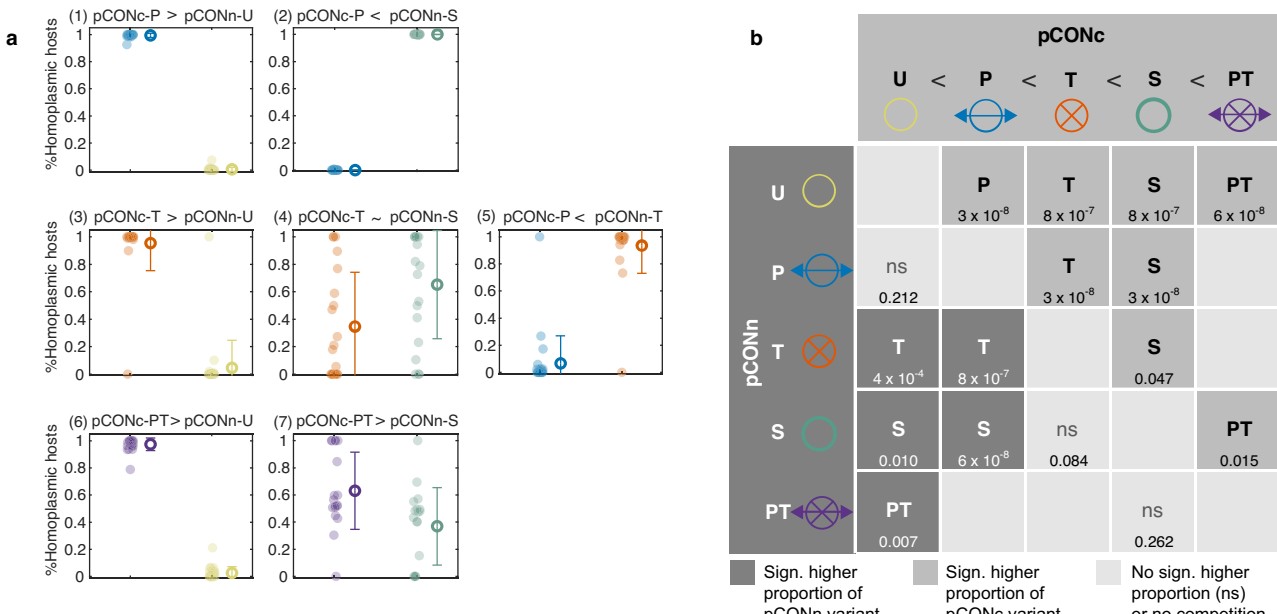

**Fig. 2 | Plasmid fitness ranking from pairwise competitions. a** Results of pairwise competitions between different pCONc and pCONn plasmid variants. Each dot corresponds to the proportion of homoplasmic hosts carrying a given plasmid variant in an individual head-to-head competition, while the mean frequency is indicated by the circle. Error bars show the standard deviation. Proportion of cell types, including heteroplasmic and plasmid-free cells, are presented in Supplementary Fig. S2) **b** Ranking of plasmid stability determinants, denoted by inequality symbols, derived from pairwise competitions of pCONn variants (rows) against pCONc variants (columns). The table cells indicate the superior competing variant having a significantly higher proportion of homoplasmic hosts at the end of the experiment. P-values below using two-sided paired sign test, $n = 25$ in all pairs of variants, except of pCONn-U vs. pCONc-PT ($n = 24$) and pCONn-S vs. pCONc-PT ($n = 20$). ns stands for not-significant (using $\alpha = 0.05$). Source data are provided as a Source Data file.

between pCONn-T and pCONc-S also reveals a higher proportion of the stable plasmid hosts compared to the TA plasmid hosts (Supplementary Fig. S3). The competitions against pCONn/c-T often resulted in heteromultimers of the competing plasmids that were observed with high frequency and were stable over time (Supplementary Fig. S4). Further comparison of plasmid fitness between the two stability systems showed that the plasmid carrying the toxin-antitoxin (pCONn-T) had a significant advantage compared to the plasmid carrying the partition system (pCONc-P) (Fig. 2a.5), which was confirmed in the reciprocal competition (Fig. S3). The similar outcomes of reciprocal competitions between plasmids carrying either a TA system or a partition system further support that the negative fitness effect of pCONn-T on the host has little influence on the results of plasmid competition. Taken together, the TA system increased the fitness of the unstable plasmid similarly to the active partition system but is superior to the partition system in a direct competition.

The competition between plasmids that harbor a combination of both systems–the active partition system and the TA system–and an unstable competitor plasmid reveals a clear advantage of plasmids pCONc-PT and pCONn-PT over the unstable plasmid variants (Fig. 2a.6; Supplementary Fig. S3). Competing the stabilized PT plasmids against the stable plasmid variant revealed a higher proportion of the pCONc-PT hosts compared to pCONn-S (Fig. 2a.7). In the reciprocal competition, the mean proportion of homoplasmic hosts of both plasmids was not significantly different (Supplementary Fig. S3). Hence, compared to the other stabilized plasmids, the plasmid harboring both stability systems had the highest fitness compared to the stable plasmid. The competitions allow for an unambiguous ranking of the plasmid stability determinants in the short-term evolution experiments. Mathematically speaking, the homogeneous relation, $>$, that indicates which stability determinant is more advantageous than another is transitive (e.g., S > T and T > U, then S > U holds). The set of pairwise competition experiments show that all stability traits PT, S, P, T were advantageous over the unstable model plasmid in one or both of the competitions.

The P trait was inferior to both the S and T traits, while the S trait had an advantage over the T trait. Finally, plasmids encoding the PT stability trait had a significant fitness advantage when competed against a stable plasmid (S) (Fig. 2b).

## A mathematical model of plasmid segregation enables a comparison among pCON stability determinants in evolutionary timescales

To predict the plasmid competitions on long time scales compared to the experimental system, we performed simulations with a model of random plasmid segregation[33], which we extended to describe the influence of toxin-antitoxin and partition systems (Fig. 3a). In models of random plasmid assortment, the probability of plasmid segregation at cell division is $2^{1-2n_c}$, which arises from the distribution of plasmid copies into daughter cells and relies on the plasmid copy number, $n_c$. Plasmid fusion or clustering of sister plasmids may decrease the number of plasmid units available for segregation, such that the probability of plasmid-free daughter cells increases. The model presented here accommodates the model plasmid properties by including the following parameters: the plasmid ability to replicate into monomeric units is parametrized by $1 - p_{\text{def}}$, where $p_{\text{def}}$ is the probability for plasmid copies to be removed from the plasmid pool available for segregation after plasmid replication. For example, if $p_{\text{def}} = 0$, all $2n_c$ copies would be available for segregation. Conversely, if $p_{\text{def}} = 1$, newly replicated plasmids would be entirely removed from the pool of independent plasmid units. The probability at which pairs of plasmid copies are actively assorted to different daughter cells in a non-random fashion (balanced assortment) is parametrized by $p_{\text{par}}$. Finally, the probability at which plasmid-free cells are killed post segregation if they divide from hosts of a plasmid harboring a TA system is parametrized by $p_{\text{eff}}$. Our model describes the population dynamics of cells with (up to) two model plasmids. While the model describes the full stochasticity of plasmid assortment to daughter cells, the host population growth is assumed to be deterministic with cell divisions in each

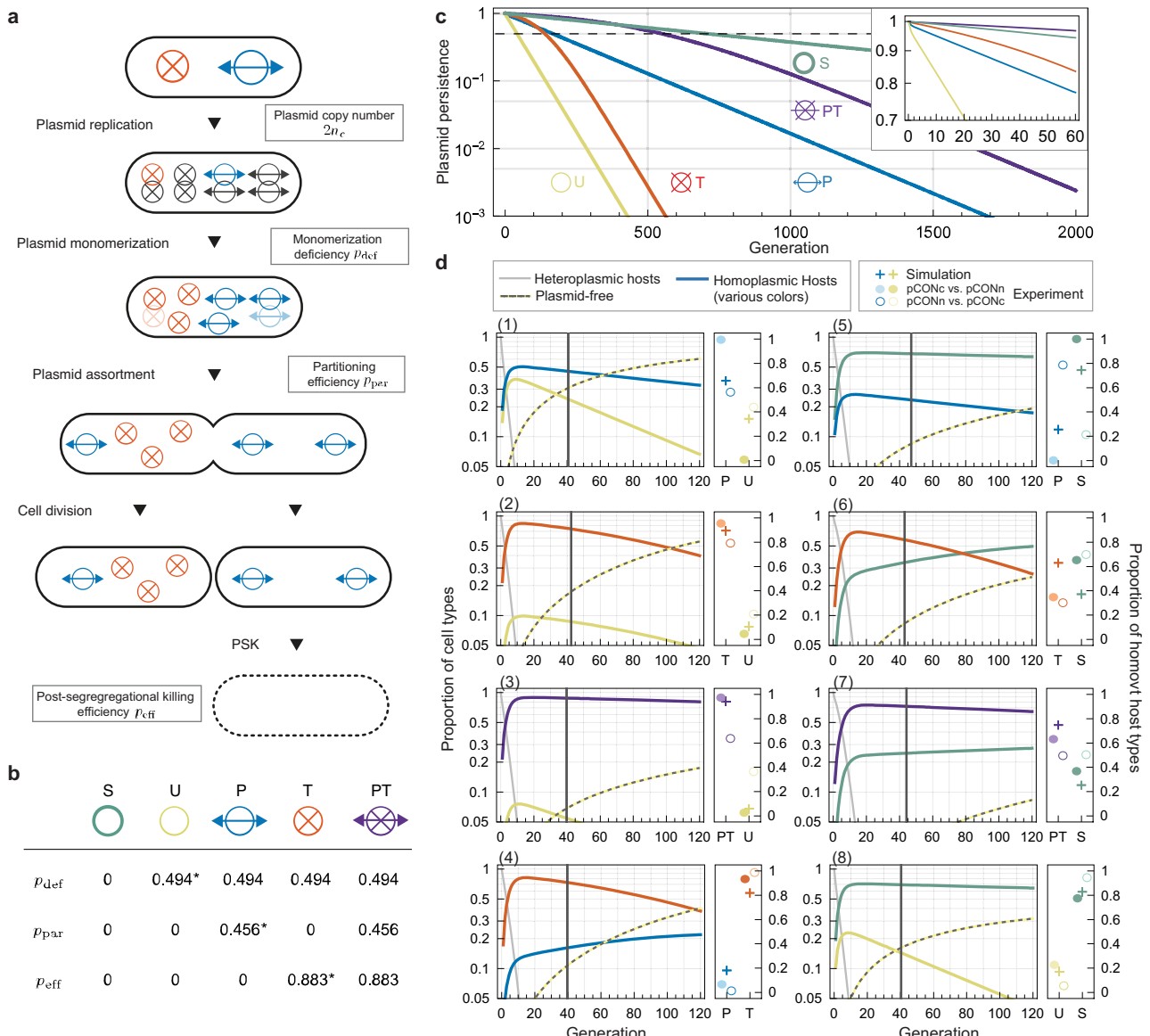

**Fig. 3 | Model of plasmid segregation and simulations of plasmid persistence and head-to-head competitions. a** Cell life cycle in the plasmid segregation model for the example of a heteroplasmic cell carrying two copies of different plasmid variants encoding a partition and toxin-antitoxin system at cell birth. The life cycle describes 1. random plasmid replication, 2. monomerization (depletion of plasmid replicates due to incomplete multimer resolution), 3. random or balanced assortment followed by cell division into two daughter cells and 4. post-segregational killing. **b** Each virtual plasmid variant, S, U, P, T, and PT, is characterized by a parameter triplet $\left(p_{\text{def}}, p_{\text{par}}, p_{\text{eff}}\right)$. The parameters $p_{\text{def}}$, $p_{\text{par}}$, $p_{\text{eff}}$ were estimated based on experimental persistence data of the respective pCONn/c variants (indicated by an asterisk, see Methods). **c** Plasmid persistence in homoplasmic cells for

various plasmid variants in the short term (inset) and long term. The dashed line indicates plasmid loss of 50% (plasmid half-life). **d** Prediction of the cell genotype dynamics in head-to-head competitions (various plots, left graph). The vertical lines highlight the generations passed at the end of the plasmid competition experiments. A comparison of simulated and experimental proportions of homoplasmic cells when experiments ended is shown (right graph). Plus markers indicate simulated proportions, while filled (and open) circles represent experimental results of the pCONc-X vs. pCONn-Y (and pCONn-X vs. pCONc-Y) competitions with the corresponding plasmid variants X, Y. Source data are provided as a Source Data file.

generation in the absence of population bottlenecks or cell death (see details in the Methods and in the supplemental text).

The virtual plasmids are defined by parametrizing $p_{\text{def}}$, $p_{\text{par}}$, and $p_{\text{eff}}$, which were estimated using the model plasmid stability data (Fig. 3c; Fig. 1c). The model does not include the effect of resistance markers (i.e., pCONn/c plasmid backbones have the same characteristics). Furthermore, plasmid-specific effect on the host fitness was not included in the model, as the negative effect of pCONn-T and pCONc-U on their host fitness had little impact on the outcome of the experimental plasmid competitions. Hence, we denote the (virtual) plasmid variants only by their stability determinant. The plasmid copy number

was set to $n_c = 5$, which is based on the mean model plasmid copy number, $\bar{n}_{\text{PCN}} = 5.09$ (Supplementary Table S1). The virtual model plasmid S is solely defined by the copy number parameter, $n_c = 5$ (Fig. 3b). For the other variants, U, P, T, and PT, we estimated the monomerization deficiency to be $p_{\text{def}} = 0.494 \pm 0.039$ (CI$_{95\%}$) from the pCONn-U/pCONc-U persistence data (Fig. 3c, see Methods). For the P and PT plasmid variants we estimated the partitioning efficiency to be $p_{\text{par}} = 0.456 \pm 0.109$ from the pCONn-P/pCONc-P persistence data. Finally, we estimated the efficiency of post-segregational killing by the T and PT variants to be $p_{\text{eff}} = 0.883 \pm 0.063$ from the pCONn-T/pCONc-T persistence data (see details in the supplemental text). The

estimated plasmid parameters were used for the simulation of long-term plasmid persistence (Fig. 3b) and pair-wise plasmid competitions (Fig. 3d).

## TA systems temporarily compensate for plasmid instability while partition systems promote long term plasmid stability

To compare the effect of plasmid stability determinants on plasmid loss dynamics in the short term and long term, we first simulated the persistence of all plasmid variants. For the stable plasmid variant (S), the segregation rate, $r^{(S)} = 2^{1-2n_c} = \frac{1}{512}$, calculated from classical random segregation models, results in an exponential decay of the plasmid host proportion, $h^{(S)}(g) = \left(1 - \frac{r}{2}\right)^g = \left(1 - \frac{1}{1024}\right)^g$, over generations, $g$ (Fig. 3c). The unstable plasmid variant (U) with an estimated monomerization deficiency of $p_{def} = 0.494$ has a ca. 16-fold higher segregation rate $r^{(U)} = \frac{1}{31.4}$. The stable plasmid variant (S) has a higher half-life (709 generations until the host cell proportion is 50%) compared to the unstable variant (U) (43 generations), as expected. Our estimation of the partitioning efficiency of the partition system encoding plasmids (P) results in an increased half-life of 170 generations. The persistence dynamics of the TA-encoding plasmid variant (T) are similar to the P-variant for the first 100 generations. When plasmid-free cells start dominating the population, the T plasmid persistence changes, however, and the host cell frequency decays exponentially at the same rate as the unstable plasmid (U). The change in the plasmid persistence trend in the model stems from two consecutive processes. Initially, plasmid-free cells arise from host cells due to random segregation; however, those are mostly killed post segregation by the toxin. The emergence of plasmid-free cells in the later phase is primarily from surviving plasmid-free cells (lacking the toxin). Hence, the model predicts that the TA system prolongs the plasmid persistence for approximately 100 generations compared to the unstable plasmid. Note that the TA system does not alter the rate of segregational plasmid loss; its effect on plasmid persistence is best understood by the resulting dynamics of host and plasmid-free cells in the population. The PT plasmid has the highest persistence with a half-life of 549 generations that is more than 3-fold compared to the P plasmid. Although the host cell proportion of the PT-variant is still substantially higher after hundreds of generations, the stable plasmid variant (S) has a higher persistence in the very long term compared to the PT-plasmid. Taken together, the stabilization systems provided to the unstable plasmid – whether acting individually or synergistically (as in the PT variant)—cannot fully compensate for the monomerization deficiency but significantly enhance the plasmid's persistence on small and intermediate time scales.

## Model predictions of plasmid competition reflect major outcomes of the experimental head-to-head competitions

To predict the outcome of intracellular pairwise competitions between plasmids with different stability determinants, we simulated the plasmid segregation dynamics in initially heteroplasmic hosts, mirroring the experimental head-to-head competitions. The simulations captured the simultaneous processes of plasmid segregation and loss (Fig. 3d). In all simulated competitions, heteroplasmic hosts rapidly declined to proportions below 10% (after 5–10 generations), while homoplasmic hosts temporarily dominate the population. Additionally, the simulations predicted a rise in plasmid-free cells to frequencies between 2% and 50% by the end of the experiments. The simulated proportions of homoplasmic hosts were similar to the empirical frequencies at the corresponding generation times with only few exceptions. In head-to-head competitions where the reciprocal experiments yielded the same outcome (i.e., excluding P vs S and PT vs. S), the simulations generally reproduced the empirical result with the exception of the competition T vs. S (Fig. 3d, plots 1–4, 6, and 8). In the T vs. S competition, the simulated proportions of the homoplasmic hosts converge during the experimental generation times and

intersect later at generation 78; thus, the S-plasmid hosts, which were found to be dominant in the experiment, are predicted to also dominate the population but only at later generation times (plot 6). Similar dynamics were observed for the competitions P vs. T (as well as for PT vs. S) with the P-variant (or S-variant) eventually dominating the population (plot 4, 7). These patterns are similar to the persistence dynamics of the respective variants in homoplasmic hosts (cf. Fig. 3c). In the competition PT vs. S, plasmid-host cells homoplasmic for the PT variant dominated for a very long period but were eventually outcompeted by the S-variant at generation 394. Taken together, the combination of partition and TA system is predicted to confer the plasmid long-term stability.

## Partition and TA systems are core functions of large plasmid types

To examine the outcome of long-term plasmid evolution on the repertoire of partition and toxin-antitoxin (TA) systems, we surveyed for their presence in 6784 plasmids from 2441 isolates of *Klebsiella*, *Escherichia* and *Salmonella* (*KES*)[34]. Plasmids in those taxonomic groups have been studied extensively[35] and are therefore expected to be well annotated. Partition systems were identified in 67% (4579) of the plasmids, all of which were classified into plasmid taxonomic units (PTUs) of large plasmids ( >19Kb). Most of the PTUs had a median of one partition system per plasmid (Fig. 4a). Multiple partition systems were observed, e.g., in plasmids within PTU-E15, -FK and -FE that can have up to five partition systems (Fig. 4a; Supplementary Data S1). Multiple partition systems may result from plasmid hybridization, e.g., as in a PTU-HI2 plasmid that harbors two functional replicons RepHI2 and RepHI2A encoding a major partition system (ParABS) and a minor system (ParMRC)[36]. The presence of both partition systems has been shown to confer highly stable segregation in a study of the related plasmid R27[37].

The survey for TA systems revealed 44 TA families in 66% (4456) of the plasmids, most of which are in PTUs of large plasmids. TA systems were rarely found in PTUs of small plasmids (except for PTU-E4 and E1). Plasmids without TA systems are exceptional among PTUs of large plasmids, examples are PTU-X3, E15, E16, and N1 (Fig. 4a, Supplementary Data S2). The maximum number of TA systems observed in a single plasmid was 11, e.g., in *Klebsiella pneumoniae* strains sampled in epidemiological surveys for antibiotic-resistant strains[38,39].

To investigate whether the apparent absence of TA systems in some large plasmids is due to limitations of our survey approach, we examined plasmids in PTU-X3 where no TA system could be identified based on currently known TA systems[40]. Plasmids in PTU-X3 (IncX3) are typically large and conjugative; many harbor antibiotic resistance *gene* $bla_{NDM-5}$ and were reported in human infections and domesticated animals[41,42]. The presence of type II TA was reported in <10% of the IncX3 plasmids[43]. A detailed examination of distant homologs to known TA systems in X3 plasmids revealed a gene coding for DinQ-like toxin (termed *dqlB*), bearing high structural similarity to a chromosomal type I TA system *dinQ-agrB* in *E. coli*[44]. An *agrB* antitoxin RNA was further identified next to a plasmid *dinQ*-like open reading frame. The genetic neighborhood of the *dqlB-agrB* includes transposable elements and is conserved across 38 (out of 116) X3 plasmids (Fig. 4b; Supplementary Data S3).

To validate the TA function of the *argB-dqlB*, we identified a relevant strain, *E. coli* EC-JS316, that harbors an IncX3 plasmid with the locus. This strain belongs to *E. coli* clone ST10 and carries plasmid pOXA-484 (IncX3) that confers the host resistance to multiple antibiotics[45]. The *dqlB* locus from that plasmid was cloned into an expression vector under the regulation of the $P_{BAD}$ promoter with or without the putative antitoxin *agrB*. Induction of the *dinQ*-like toxin expression led to growth inhibition of the host, while induction of the complete *agrB-dqlB* expression showed no signs of growth inhibition (Fig. 4c). Our results thus confirm a TA function of the *agrB-dqlB* locus

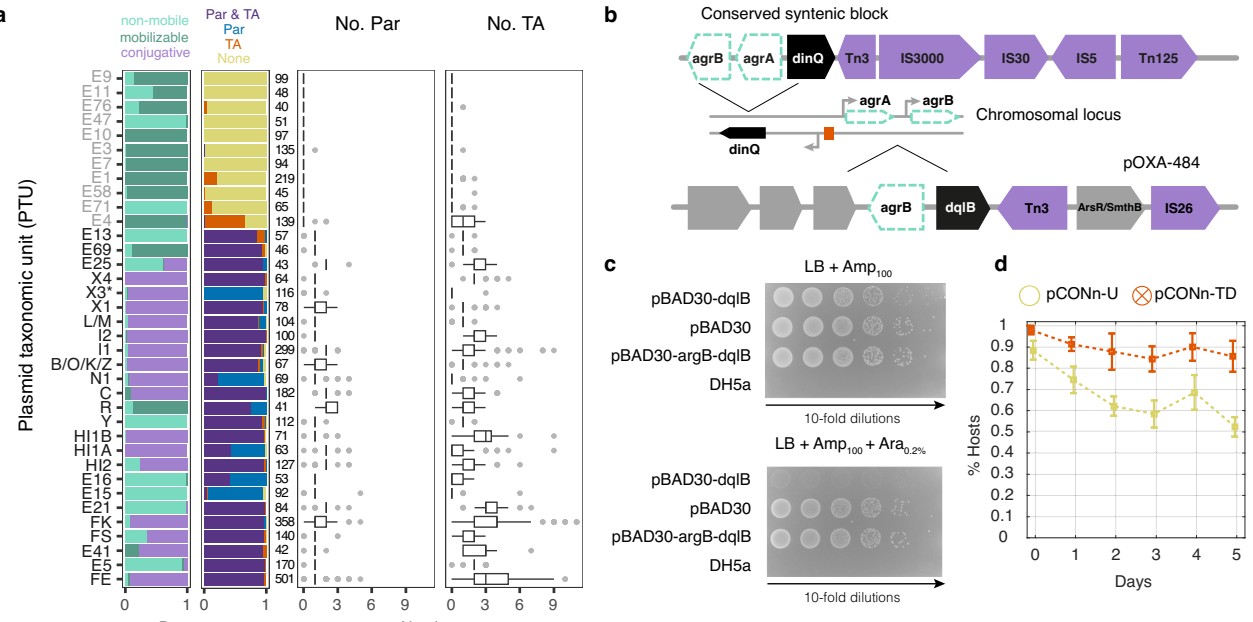

**Fig. 4 | Large plasmids likely harbor both partition and TA systems. a** Presence of active partition (Par) and toxin-antitoxin (TA) systems in *KES* plasmid taxonomic units. PTUs of small plasmids (<19Kb) are written in gray; PTUs of large plasmids (≥19Kb) are written in black. Horizontal stacked bar graphs (first column) show the fraction of plasmid mobility in the PTU, while bar graphs (second column) show the systems' presence in plasmids, and box plots show the distribution of system instances per plasmid. Boxes (whiskers) indicate the interquartile range (1.5 x IQR) and vertical lines show the median. The number of plasmids in each PTU is indicated by the numbers left of the box plots. X3* are presented here as devoid of TA system, i.e., prior to our validation of the *dinQ*-like system. **b** Structure of a conserved syntenic block including *dinQ*-like gene in X3 plasmids. The chromosomal *dinQ* in *E. coli* is presented together with the *agrB-dqlB* locus in plasmid pOXA-484

in *E. coli* strain EC-JS316. **c** Toxicity assay of the *agrB-dqlB* locus from pOXA-484. Growth of *E. coli* DH5α cells carrying either no plasmid, the empty vector control plasmid (pBAD30), the *argB-dqlB* locus from plasmid pOXA-484 in *E. coli* EC-JS316 (pBAD30- *argB-dqlB*), or the *dqlB* carrying plasmid (pBAD30-*dqlB*). The samples were plated on medium selective for ampicillin resistance (LB+Amp$_{100}$) and selective plates supplemented with Arabinose (LB+Amp$_{100}$ + Ara$_{0.2\%}$). The expression of *dqlB* locus is induced in the presence of Arabinose via P$_{BAD}$. The presented results are representative of three independent replicates. **d** Stability of plasmid pCONn-TD comprising the *agrB-dqlB* locus from plasmid pOXA-484 as evaluated in a serial transfer experiment under non-selective conditions (*n* = 6 replicates with error bars corresponding to standard deviation). Source data are provided as a Source Data file.

in pOXA-484. To further validate the plasmid stability function of the *agrB-dqlB* TA system, we cloned the native pOXA-484 *agrB-dqlB* locus into the pCONn backbone (termed pCONn-TD). Quantifying the persistence of pCONn-TD showed an increase in plasmid stability compared to pCONn-U (Fig. 4d). We conclude that the DinQ-like TA system has a role in the persistence of IncX3 pOXA-484-like antibiotic resistance plasmids, in agreement with recent studies reporting similar results[46,47]. We conclude that large conjugative plasmids likely harbor TA systems, even if those cannot be identified by sequence similarity to the so far recognised systems.

## Discussion

Extrachromosomal and mobile genetic elements are frequent across all domains of life. The replication and segregation of genetic elements that integrate into the host chromosome is secured by the basic cellular process; examples are transposons in bacteria (e.g., Tn7[48]) and insertion elements in Archaea (e.g., ISC1217[49]), starships in fungi[50], P-elements in plants[51] and SINEs in human[52]. In contrast, the persistence of autonomously replicating elements largely depends on their ability to persist within the host lineage. Studies of extrachromosomal genetic elements that evolved into an integral component of the host genome suggest that such elements harbor mechanisms for the synchronization of their replication with the host chromosome and their segregation with cell division. Examples are the secondary chromosome in the bacterium *Vibrio cholerae*[53]), accessory chromosomes in fungi[54], B-chromosomes in plants and animals[55] as well as the eukaryotic organelles[56]. Nonetheless, the specific molecular mechanisms at the origins of such elements often remain poorly understood. Here, we provide empirical insights and theoretical predictions into the

evolutionary success of extrachromosomal elements depending on their persistence strategy with the example of prokaryotic plasmids.

Our study shows that molecular mechanisms that increase plasmid stability are likely to increase also the plasmid evolutionary success (i.e., fitness). When the difference in plasmid stability determinants is large, such as between the unstable and stabilized pCON variants, the outcome of the competition favors the plasmid with higher stability. Predicting the evolutionary trajectory in competitions between plasmids having a similar stability is, however, more challenging. Our approach of intracellular competitions supplies a methodological solution for the comparison of plasmid fitness, also between plasmids having a similar stability. The performed competitions reveal that the stabilized plasmids outcompeted the unstable plasmid, hence both partition and TA systems have the potential to increase the fitness of plasmids that are impaired in their replication and multimer resolution (such as the pCON backbone[30]). At the same time, the stabilized plasmids had lower fitness in the competition against the stable pCON$n/c$ -S, hence, active partition and TA systems are not superior in comparison to plasmid traits that enable the plasmid to effectively replicate into monomeric entities. The combination of both partition and TA systems was revealed to be superior in the short-term experiments, even above the highly stable plasmid. This result may be explained by the contribution of a partition system to reduce the frequency of segregants (plasmids-free hosts)[21], which is beneficial if the toxin effectivity is low (e.g., due to the toxin instability[57]).

The mathematical model reveals systematic differences in the influence of partition and toxin-antitoxin systems on plasmids' long-term persistence. Partition systems slow down the plasmid decay in

the population by reducing the segregation rate, while toxin-antitoxin systems only delay plasmid loss via their negative effect of post-segregational killing on newly generated plasmid-free cells. Since the TA system *hok/sok* used in our experiments is a highly efficient toxin-antitoxin system, it allowed only a few plasmid-free lineages to arise in the initial population of plasmid hosts and, thus, effectively suppressed plasmid loss. Toxin-antitoxin systems can be most efficient in stabilizing the plasmid compared to the active partition system when plasmid hosts make up (almost) the entire population. Our findings, thus, suggest that TA systems play a role in plasmid stability after periods of strong positive selection for plasmid hosts. When the proportion of plasmid host cells becomes small, plasmid-free cells are mainly the offspring of plasmid-free cells. In such a scenario, the frequency of a plasmid harboring a TA system decays at nearly the same rate as the unstable plasmid because post-segregational killing only affects a small proportion of the plasmid-free subpopulation. For plasmids encoding both TA and partition system (PT), the partition system additionally stabilizes the plasmid with the result that low plasmid-host proportions are reached later, thus the TA system remain efficient for additional generations, which results in the very successful strategy of the PT-variant and is in line with previous experimental result[21]. Taken together, our mathematical model provides a new framework to evaluate the interplay of partition systems, toxin-antitoxin systems, and plasmid copy number on the plasmid persistence.

Our results offer insights into the allele dynamics of plasmid stability traits. The starting point in our experimental and model setups is aimed to mimic allele segregation post-balancing selection event (i.e., starting from similar frequencies of both alleles). Heteroplasmy of plasmid alleles can be maintained under rapidly alternating selection for both alleles, as has been demonstrated for antibiotic resistance alleles[58]; the maintenance of heteroplasmy requires, however, that the selection for both alleles is strong[59]. The experimental plasmid competitions presented here show that the segregation of plasmid variants with (on average) five copies is rapid (Figs. 1, 2). The simulated plasmid competitions show that the competition outcome in later generations is determined by the dynamics of homoplasmic plasmid-hosts (Fig. 3d). A higher plasmid copy number is expected to result in extended states of plasmid heteroplasmy (termed also heterozygosity window[60]), as demonstrated for allele dynamics of a plasmid having ca. 15 copies per cell[61]. Random segregation of plasmid alleles during cell division (termed segregational drift[62]) may lead to a rapid loss of plasmid alleles even if they confer a fitness advantage to the host[33]. The mode of plasmid replication and segregation is furthermore expected to affect the length of the heteroplasmy state[60,63]. However, empirical data on the mode of plasmid replication and segregation remains lacking, with only a few exceptions[33], hence the presence of mechanisms for maintenance of plasmid heteroplasmy (if any) remains unknown. Alternatively, hints on the effect of copy number and mode of segregation on the pace of plasmid allele dynamics may be gained from observations made for other types of extrachromosomal genetic elements. Example is a recent study that investigated allele dynamics of eukaryotic organelles in clonal plants using a mathematical model similar to the one presented here. Calibrating the model with deeply sequenced samples led to the conclusion that heteroplasmy is different between the mitochondrion and plastid due to different modes of segregation, and is furthermore rather rare[64]. Taken together, we conclude that plasmid heteroplasmy is expected to be rare and the fate of plasmid variants is most likely determined by the dynamics of plasmid persistence in homoplasmic hosts.

Our experimental and theoretical results show that the combination of post-segregational killing mechanism (TA) and active partitioning systems is superior over all other plasmid persistence strategies (under conditions that are non-selective for the plasmid presence). The prevalence of such combinations in large plasmids, further supports our observations (Fig. 4a). Large plasmid size is typically associated with self-transmissibility or mobility, where the presence of conjugation and mobility modules contributes to the increase in plasmid size[65]. Indeed, not all large plasmids in our analysis are conjugative (Fig. 4a). Nonetheless, recent studies suggest that large plasmids in several enterobacteriaceae PTUs annotated as mobilizable or non-mobile evolved from conjugative plasmid ancestors[66]. For example, PTU E5 that comprises mostly non-mobile plasmids was inferred to be related with PTU FE that comprises mostly conjugative plasmids. Plasmids in PTU E5 comprise multiple pseudogenes, including non-functional copies of conjugation genes, and are found in a narrow host range, both are characteristics of domesticated plasmids[67]. The prevalence of non-functional transfer (*tra*) gene copies in other PTUs of large plasmids in *Escherichia* and *Salmonella*[67] suggests a common evolutionary origin, with PTUs of large mobilizable or non-mobilizable plasmids having been derived from conjugative plasmid ancestors.

The prevalence of partition and TA systems in conjugative and mobilizable plasmids indicate they are important determinants of plasmid fitness regardless of plasmid mobility. Indeed, plasmid mobility is commonly considered a plasmid persistence mechanism that is expected to increase plasmid fitness. Nonetheless, several studies demonstrate that vertical inheritance often dominates plasmid persistence within the population, especially under conditions that limit the frequency of conjugation or where fitness costs of plasmid carriage are mitigated[68–71]. Compared to conjugative plasmids, our model plasmids are rather small (2.7–5.3 Kb; Fig. 1b), however, they share a similar copy number of ca. 5 copies per cell with large *Escherichia* plasmids[14]. Hence, we consider the plasmid copy number a key determinant of our results and predictions. Our simulation of long-term plasmid persistence predicts that the most successful stability determinants do not ensure infinite persistence and highlight important systematic differences in the effect of TA and partition systems on plasmid evolutionary success. Indeed, plasmids in many of the large plasmid PTUs may harbor and disseminate antibiotic resistance genes[34,72], hence their long-term persistence is likely promoted in addition by periodic selection events (and to dispersal across host populations) (e.g., as shown for pOXA-48[73]). We conclude that the combination of TA and partition systems being the most effective strategy for the persistence of low-copy extrachromosomal elements whose segregation with the host is unstable.

## Methods

### Bacterial strains and culture conditions

Bacterial cultures were grown routinely at 37 °C with shaking or on Lysogeny Broth (LB) agar plates unless stated otherwise. For selective conditions, the following concentrations were used: kanamycin 10 µg/mL; chloramphenicol 10 µg/mL; trimethoprim 125 µg/mL; tetracycline 10 µg/mL. The *E. coli* strain MG1655 *recA*::*tetA* derived from the *E. coli* strain MG1655 (DSM No. 18039, German Collection of Microorganisms and Cell Cultures, DSMZ) was used as plasmid host in all competition experiments. The construction of the model plasmids was performed with *E. coli* DH5α[74] as a cloning host. Additionally, the toxicity assay was also conducted in DH5α. All primers used in this study are listed in supplementary Table S1.

### Construction of strains and plasmids

Previously we observed that model plasmids of the pCON backbone tend to form heteromultimers, and those multimers can be stable over time[6]. Since plasmid multimers are rarely observed in *Escherichia* strains[75], we consider the presence of pCON heteromultimers a specific property of the pCON backbone in our experimental setup. To decrease the effect of such multimers on the results of plasmid competitions, we performed the plasmid competitions in a *recA* knockout mutant. To generate the *E. coli* MG1655 *recA*::*tetA* strain the parental strain MG1655 was first transformed with the helper plasmid pTKRED[76],

harboring genes encoding the λ-Red enzymes for recombineering. The *tetA* gene, along with its promoter $P_{lacIQ1}$ was amplified using pTKLP-*tetA* as template. For the amplification of *tetA* with its promoter, the primers NH300 and NH301 were used along with Phusion polymerase. After acquiring the MG1655 strain containing pTKRED a liquid culture with IPTG (2 mM) was inoculated. Electrocompetent cells were generated after an overnight incubation at 30 °C. The electrocompetent cells were then transformed with the purified PCR product. The transformed cells were incubated at 30 °C and plated onto LB agar plates with tetracycline. The transformants were cured of pTKRED by incubation at elevated temperature (37 °C).

A trimethoprim-resistant strain used for the assessment of plasmid fitness effects was generated through triparental mating of the host strain, *E. coli* MG1655 *recA::tetA*, with the two *E. coli* strains MFD*pir* pGP-Tn7-Tp[77] and MFD*pir* pTNS2[78]. The three strains were incubated overnight in liquid culture at their respective optimal temperature. Following this, the three strains were washed twice by centrifuging for 5 min at 3000 rpm and resuspension in Phosphate Buffered Saline (PBS). Afterwards, the cells were resuspended in 500 μL of PBS and mixed in a 1:1:1 ratio based on their $OD_{600}$. The strain mix was spotted onto a non-selective agar plate with diaminopimelic acid (DAP) and incubated at 30 °C overnight. After the mating, the conjugation spot was resuspended, the cells plated onto selective agar plates without DAP. This step hinders the growth of the MFD*pir* donor strain due to its requirement for DAP, as well as the replication of plasmids pGP-Tn7-TP and pTNS2 in the recipient strain due to their requirement for *pir*-encoded replication initiation protein π (PI), which is encoded only in the MFD*pir* strain, leaving only the plasmid-free recipient strain alive on the plate.

For the construction of the model plasmids, the plasmids pCON (termed here pCONn) (GenBank accession number MK697350)[30] and pCON2 (termed here pCONc)[6] were used as the plasmid backbone. The plasmids pCONn-U and pCONc-U were generated via Gibson Assembly®. Here, the plasmids pCONn and pCONc were amplified using the primer pair #104/#105. As an insert, a sequence of the plasmid pLC62 was amplified using the primer pair #101/art2_F(JI). This insert contains a non-coding DNA sequence and an ampicillin resistance gene (total 2258 bp).

The plasmids pCONn-P and pCONc-P were generated by inserting the *parA* locus of plasmid R1 into the respective model plasmids. The *parA* locus, which contains *parM*, *parR* and *parC*, which constitute an active partitioning system, was amplified using the primer pair #98/#106. The PCR product and the plasmids pCONn and pCONc were digested using HindIII, and the insert was cloned into the plasmid backbone. The construction of pCONn-T and pCONc-T was done via Gibson Assembly® (NEBuilder® protocol; New England Biolabs) using the primer pair #99/#100 to amplify the *parB* locus of plasmid R1, containing the *hok/sok* toxin-antitoxin system. The primer pair #117/#118 was used to amplify plasmid pCONn and plasmid pCONc. To construct the plasmids pCONn-PT and pCONc-PT, the plasmids pCONn-P and pCONc-P were amplified using the primer pair #114/#115. The *parB* locus of plasmid R1 was amplified using the primer pair #112/#113. The PCR products were then used for a Gibson Assembly®.

The donor of the *agrB-dqlB* locus was the self-transmissible pOXA-484 plasmid, a previously described IncX3 plasmid[45]. The locus was cloned into plasmid pBAD30 under the control of a $P_{BAD}$ promoter[79]. The construct pBAD30-*dqlB* carries the *dqlB* gene. Primers TF21/22 were used for the *dqlB* insert, while the backbone was digested using the restriction enzymes XbaI and EcoRI (Supplementary Data S4). The construct pBAD30-*argB-dqlB* incorporates the complete *agrB-dqlB* locus downstream of the $P_{BAD}$ promoter. Primers TF57/58 were used for the complete locus insert, and pBAD30 was digested with Eco53kI (Supplementary Data S4). The plasmid pCONn-TD was constructed from pCONn with the complete *agrB-dqlB* locus incorporates in the same locus as the other plasmid stability modules (Fig. 1b). The plasmid pCONn-TD was constructed using TF53/54 for the complete *agrB-dqlB* locus insert and TF49/50 for the backbone (Supplementary Data S4). All pBAD30-derived plasmids, as well as pCONn-TD, were constructed via Gibson Assembly®.

## Preparation of electrocompetent cells
The electrocompetent cells were prepared according to the protocol described by Dower et al.[80].

## Transformation
The transformation during plasmid construction was performed with a heat-shock protocol using chemically competent cells[81]. Competent cells aliquots were thawed, mixed with the DNA and incubated on ice for 30 min. The transformation tubes were shifted to a water bath at 42 °C and incubated between 45–90 s, followed by 2 min rest on ice. The transformation mix was then transferred into fresh LB medium and incubated for 1 h at 37 °C and 200 rpm. After the outgrowth period, appropriate dilutions and volumes were plated on selective LB agar plates and incubated overnight.

The introduction of model plasmids into the naïve host population was performed with electroporation. The electrocompetent cells were mixed with the respective amounts of plasmid DNA and transferred into cuvettes with a gap of 1 mm. The cells were electroporated at 1350 V, 25 uF and 600 Ω using the BioRad Genepulser Xcell (Bio-Rad Laboratories Ltd., CA, USA). The transformation mix was transferred into 1 mL of fresh LB medium and incubated for 1–3 h at 37 °C with 200 rpm. The cultures were then either exposed to positive selection in liquid media or plated on selective LB agar plates.

## DNA sequencing and plasmid copy number
In order to estimate the plasmid copy number (PCN), host populations were grown in liquid media with either Km or Cm for 16 h. The bacterial culture had therefore reached the stationary phase. From the cultures, chromosomal (cDNA) and plasmid DNA (pDNA) were extracted using the Promega WIZARD (Promega, WI, USA) DNA extraction kit as instructed by the manufacturer. The acquired DNA sequenced by Illumina (Eurofins Scientific SE, Lux) to an average sequencing depth of 469.31 cDNA and 2698.64 pDNA (Supplementary Table S1). This allowed us to calculate the average PCN for the plasmid backbone. The analysis of the sequencing data was done using Galaxy version 24.1[82]. Plasmid sequencing for validation was done by extracting plasmids using the Monarch Miniprep Kit (New England Biolabs, MA, USA) followed by Whole Plasmid Sequencing or Sanger Sequencing by Eurofins.

## Quantification of plasmid loss
For the plasmid stability assays, a population of plasmid host were incubated overnight incubation at 37 °C and 200 rpm in liquid media under selective pressure with the respective antibiotic. If the cells were taken from a cryostock, an additional day of growth on a selective plate was added prior to the preadaptation in liquid medium. Following the preadaptation, the $OD_{600}$ of the cultures was used to calculate the volume required for inoculation of fresh medium to an OD of 0.01. The ratio of hosts within the population at $t_0$ was determined by differential plating after inoculation of the liquid culture. Every 24 h for five days, i.e., $t_0$ to $t_{120}$ the proportion of hosts was determined by plating on LB plates and selective plates containing antibiotics according to the tested plasmid: 10 ng/μL Km for pCONn variants; 10 ng/μL Cm for pCONc variants. In each day, fresh LB liquid media was inoculated with the previous culture, using a dilution of 1:100.

## Plasmid fitness effect on the host
The plasmid-carrying strains and the plasmid-free strain were competed against a plasmid-free strain, which is isogenic except for a TpR gene supplying resistance to trimethoprim. Bacterial cultures were

first preadapted in liquid media for three days. The preadaptation was done with and without antibiotics, depending on whether the respective strain carries a plasmid. The cultures were transferred to fresh liquid LB media every 24 h with dilution of 1:100. After the pre-adaptation, the strains were diluted 1:10, and the OD600 of the strains was measured and adjusted to 0.1. The competing strains were used to inoculate 1960 μL of fresh LB media, with 20 μL from strain 1 and another 20 μL from strain 2. The competition mix was then divided into two 1000 μL volumes, where one was incubated overnight and the other was used to determine the frequency of competing strains at the starting point $t_0$. The strains' frequency at $t_0$ and $t_{24}$ was determined by plating the competition mix in appropriate dilutions onto non-selective LB agar plates, LB agar plates with either Km (10 ng/μL) or Cm (10 ng/μL) and LB agar plates with Trim (125 ng/μL). The relative fitness ($w$) was calculated using the Malthusian parameter formula and compared between plasmid genotypes with two-sided Wilcoxon test.

## Intracellular plasmid competitions

In the intracellular head-to-head plasmid competition, an *E. coli* MG1655 *recA*::*tetA* host population was electroporated and transformed with the two competing plasmids, with 1 ng of plasmid DNA (or more) each, at the same time. This was followed by 3 h of incubation in non-selective liquid LB medium to allow for plasmid establishment in the host. After the outgrowth period, antibiotics were added to a concentration of 10 ng/μL for Km and 3 ng/μL for Cm. The liquid culture was then incubated overnight for -16 h. The culture was then diluted and plated on double selective media (Km 10 ng/μL, Cm 10 ng/μL) and incubated at 20 °C for 3 days to ensure that only hosts carrying both plasmids survive, while keeping the growth rate minimal. Five out of the colonies that grew on the double selective plate (termed initial colonies) were excised and resuspended in PBS (see also Fig. 1a). The suspended initial colonies were plated on non-selective agar LB plates that were incubated overnight at 37 °C. After the incubation period, five colonies per initial colony were excised and resuspended in PBS. The frequency of hosts and plasmid-free cells was quantified by plating on non-selective LB agar plates and selective LB agar plates with either Km (10 ng/μL), Cm (10 ng/μL) or both antibiotics. Additionally, the presence of competing plasmids was validated in the initial colonies in a similar manner.

## Model simulation

We simulate the population dynamics of plasmid hosts and plasmid-free cells using a model of random segregation extended for incomplete plasmid monomerization post replication, active partitioning and post segregational killing (induced by partition systems and toxin antitoxin systems respectively). The model describes the population dynamics of bacterial cells characterized by the abundances $i_{p_1}$ and $i_{p_2}$ of plasmid copies of two different plasmid variants, referred to by $p_1$ and $p_2$, at cell birth. Plasmids copies replicate following a Polya urn scheme until the cell harbors $2n_c$ plasmid copies, i.e. plasmids are randomly selected for replication from the cell's plasmid pool and replicates are retrieved into the pool before the next replication event, which results in $J^{\text{rep}}_{p_1}$ plasmid copies of variant $p_1$ and $J^{\text{rep}}_{p_2} = 2n_c - J^{\text{rep}}_{p_1}$ of variant $p_2$ in the cell; note that $J^{\bullet}_{p_1}, J^{\bullet}_{p_2}$ here and below are random variables. For the stable plasmid variant, $p_{1,2} = S$, plasmid copies monomerize (optimally) leaving $J^{\text{mon}}_S = J^{\text{rep}}_S$ plasmid units that segregate randomly at cell division. For the other plasmid variants, $p_{1,2} \in U, P, T, PT$, we model suboptimal monomerization by removal of from the pool of the (new $J^{\text{rep}}_{p_1} - i_{p_1}$) plasmid replicates by binomial sampling with parameter $p_{\text{def}}$ (monomerization deficiency) leaving $J^{\text{mon}}_{p_{1,2}}$ copies (of the respective plasmid variants) before cell division. For the plasmid variant harboring a partition system, $p_1 \in P, PT$, plasmids form $\lfloor J^{\text{mon}}_{p_1}/2 \rfloor$ pairs and plasmid copies of pairs are independently successfully partitioned (divided) at cell division at probability $p_{\text{par}}$, i.e., they are distributed to different daughter cells, while the

remaining copies are randomly distributed, resulting in $J^{\text{div}}_{p_1}$ copies in one (of the two) daughter cells (see supplemental text for all probability distributions). The mathematical derivation of the probabilities $p_{i \to j}$ that a cell harboring $i_{p_1}$ and $i_{p_2}$ plasmid copies of variant 1 and 2 at cell birth, respectively, denoted by $i = (i_{p_1}, i_{p_2})$ gives rise to a daughter cell of type $j = (j_{p_1}, j_{p_2})$ with $j_{p_1}$ and $j_{p_2}$ plasmid copies of variant 1 and 2 allows us to compute the expected number of $j$-type cells produced at cell division of an $i$-type cell. For plasmid variants encoding at toxin-antitoxin cells, $p_{1,2} \in T, PT$, cells may die from post-segregational killing. If the daughter cells lacks the plasmid ($j_{p_{1,2}} = 0$) while the parental cell carried a plasmid encoding the TA system ($i_{p_{1,2}} > 0$, see supplemental text for the case that both plasmid variants, $p_1$ and $p_2$, carry the TA system, which we do not examine in the results shown), cells only survive with a small chance $p^{\text{surv}}_{i \to j} = 1 - p_{\text{eff}}$ due to post-segregational killing escape. We defined the expected number of $j$-type cells that are produced at cell division of an $i$-type cell,

$$2p_{i \to j}p^{\text{surv}}_{i \to j} =: A_{ji}, \tag{1}$$

which were used to compute the cell-type proportions after $g$ generations,

$$\boldsymbol{x}(g) = \frac{\mathbf{A}^g \boldsymbol{x}_0}{|\mathbf{A}^g \boldsymbol{x}_0|}, \tag{2}$$

where $\boldsymbol{x}$ is a vector containing the proportions of all cell-types, $\mathbf{A}$ is a matrix with elements $A_{ji}$, and the vector $\boldsymbol{x}_0$ describes the initial population state ($x_{0,i} = 1$, for $i = (i_{p_1}, i_{p_2}) = (1, 1)$ for initial populations of heteroplasmic hosts and for $i = (i_{p_1}, i_{p_2}) = (1, 0)$ for populations of homoplasmic hosts). Mathematical derivations and details on the estimation of the parameters can be found in the supplemental text. Simulations were performed using Mathematica® (V. 14).

## Annotation of active partition systems and toxin-antitoxin systems in plasmid genomes

For the survey of active partition systems and toxin-antitoxin systems, we used a previously established *KES* dataset of enterobacterial genomes. The *KES* dataset includes 1114 chromosomes and 3098 plasmids from *Escherichia*; 755 chromosomes and 2693 plasmids from *Klebsiella*; 572 chromosomes and 993 plasmids from *Salmonella* (RefSeq database; version 01/2021). Protein coding genes in the *KES* genomes were clustered into 32,623 gene families as previously described[34]. Briefly, reciprocal best hits (RBHs) of protein sequences between all replicon pairs were identified using MMseqs2[83] (v.13.45111, with module easy-rbh applying a threshold of E-value ≤ 1 × 10^{-10}). RBHs were further compared by global alignment using parasail-python[84] (v. 1.2.4, with the Needleman-Wunsch algorithm). Sequence pairs with ≥ 30% identical amino acids were clustered into gene families using a high-performance parallel implementation of the Markov clustering algorithm[85] (HipMCL with parameter −abc -I 2.0).

We first collected NTPase and CBP protein sequences, associated with type I, II, III, and a novel type (StbA) partition systems from the literatures[12,86,87]. These sequences were aligned against the 32,623 clusters of homologous gene families using BLAST[88], followed by manual inspection of gene location and orientation, we kept only the neighboring pairs of NTPase and CBP encoded on the same strand (except StbA system). Our survey resulted into 34 partition systems in the plasmids (Supplementary Data S1). Note that our search did not include the *par* sites (centromere-like DNA sequence). Active partition systems were exclusively found in plasmids with size larger than 19 kbp, with exceptions of seven plasmids smaller than 19 kbp, which were identified as putative truncated assemblies after BLAST searches against NCBI nr database.

Utilizing the Toxin-Antitoxin Database (TADB) and the tool TAfinder (version 2.0)[40] we identified TA systems belong to 44 families in 4456 plasmids (a full list of TAs is in Supplementary Data S2). After organizing TA system-carrying plasmids into PTUs, we manually searched for toxin gene annotation in those PTUs where no TA system was detected. A *dqlB* gene encoding DinQ-like toxin was annotated after a recent RefSeq update, *dqlB* shares 41.2% global identity with the amino acids sequence of DinQ toxin in *E. coli* K-12 strain MG1655 chromosome. ESMfold[89] revealed the same helix structure of the two proteins, TM-align[90] further showed their highly structural similarity with the TM-scores 0.96 and 0.85. We used BLAST against the Rfam database to predict the two small RNA sequence, *agrA* and *agrB*, in the potential *dqlB* locus on PTU-X3 plasmids. This resulted in a prediction of *argB* antisense RNA locus in the neighborhood of *dqlB* but failed in predicting an *agrA* locus.

### Reporting summary

Further information on research design is available in the Nature Portfolio Reporting Summary linked to this article.

## Data availability

The data underlying this article are available in the article and in its online supplementary information, supplementary data and supplementary source data. Sequencing data of plasmid copy number estimates generated in this study are available at ENA under accessions: PRJEB90830. Detailed descriptions of model plasmids established in this study are available at opendata@uni-kiel under https://doi.org/10.57892/100-257. Source data are provided with this paper.

## Code availability

The code underlying the model simulations presented in Fig. 3 are available at Zenodo https://doi.org/10.5281/zenodo.14975655.

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

## Acknowledgments

We thank Stephan Göttig from the University Hospital, Goethe University Frankfurt am Main, for contribution of *E. coli* EC-JS316 isolate. We thank Lisa Hartmann for critical comments on the manuscript. We thank Susanne Landis for graphical illustrations. This research was supported in part through high-performance computing resources available at the Kiel University computing center. German Science Foundation; RTG 2501 TransEvo, grant number: 456882089. European Research Council; pMolEvol, grant number: 101043835.

## Author contributions

J.E., N.H. and T.D. conceived the study and designed plasmid competitions. J.E. performed plasmid competitions and analyzed their results. J.E., M.S. and T.D. conceived model simulations. M.S. designed and performed model simulations. Y.W. and T.D. conceived the plasmid comparative genomes. Y.W. designed and performed plasmid comparative genomics and detection of TA systems. T.F. and N.H. designed the TA system validation experiments. T.F. performed TA validation experiments and analyzed their results with assistance from NH. All authors wrote the manuscript.

## Funding

## Competing interests

The authors declare no competing interests.
