## [Transparent Peer Review file · Nature Communications]

The combination of active partitioning and addiction mechanisms is most advantageous for low-copy plasmid fitness

Corresponding Author: Professor Tal Dagan

Version 0:

Reviewer comments:

Reviewer #2

(Remarks to the Author)

Overall, we're pleased with the authors' response. The manuscript is now clearer and better framed within the existing literature. The authors have effectively addressed concerns regarding novelty and statistical issues.

However, some issues still require attention. The authors state that direct comparisons between the PT plasmid and the P or T plasmids could not be conducted due to the risk of recombination at identical loci. Nevertheless, all pCON variants share the same backbone. If recombination is truly frequent enough to preclude PT vs. P or PT vs. T competitions, it raises the question of how it was ruled out as a confounding factor in the other experiments. A brief mention of the potential effect of recombination on the observed results should be sufficient to address this issue.

Furthermore, the abstract indicates that the combination of partitioning and TA systems is more effective than either strategy alone, implying that this comparison was directly tested. In reality, the advantage of the combined PT system was inferred from modeling and indirect competition results, rather than being experimentally demonstrated through direct PT vs. P or PT vs. T competitions. While the inference is well supported, the abstract could be phrased more cautiously to acknowledge this limitation.

We anticipate these issues to be easy to solve, and we congratulate the authors on a strong and well-executed study.

Reviewer #3

(Remarks to the Author)

I thank the authors for their responses to my comments. All responses were reasonable and take away most of my concerns. With respect to my comment 4 about large (often conjugative) plasmids typically having both plasmid persistence mechanisms (TA and partitioning), I can see that it may be too simple to perform the statistical analysis I suggested given the effect of conjugation genes on plasmid size. However, it may still be informative to the reader to mention -- and possibly briefly discuss -- this association with plasmid size.

Reviewer #4

(Remarks to the Author)

Response to referees' comments on Effe et al. 2025.
We supply the manuscript tracked changes version in the submission files.

Reviewer Expertise:

Referee #1: Toxin-antitoxin systems
Referee #2: plasmid biology
Referee #3: plasmid biology, AMR, evolution

Reviewers Comments:

Reviewer #1 (Remarks to the Author):

Effe et al. present an interesting analysis on the genetic factors that affect long-term persistence of plasmids in bacteria. They establish a carefully designed well-controlled experimental system to test which of partition system, TA system, or both is the most efficient strategy for maintaining a plasmid. Next, they used the collected parameters to develop a mathematical model for the long term persistence of plasmids using these genes. They report that TA systems temporarily compensate for plasmid instability while partition systems promote long term plasmid stability. Then they surveyed the presence of TA and Par systems in thousands of plasmids from 3 well studied pathogenic enterbacterial genera (KES). And finally, they validated a new TA system carried on IncX3 and showed that the new system increases the stability of the plasmid in their original experimental system.

Response: we thank the referee for the summary of our work and the kind words.

--o

Major comments:

1. I believe the results. The experimental system is well-controlled. The mathematical model should deserve more explanation for people without the right background. In a perfect world the model could have been experimentally validated. After all there is clear difference after a few dozens of *E. coli* generations. The discovery of a new TA system does not contribute a lot to the paper in my view. In general, I am not sure this work has the right impact for Nature Microbiology. It seems like an important work in plasmid molecular evolution field but I am not sure that it will be of high interest to broad microbiology readership.

Response: These are valid comments.

a. we modified Fig. 3a to include additional explanations on the model by showing at which stage of the life cycle each parameter has an impact. Please note that the model is described in much more detail in the supplementary material.

b. Our manuscript includes two approaches for the validation of the model results. Note that the model is calibrated only with the empirical plasmid stability data – not the plasmid competition results. The results of the model are compared to the results of the plasmid competitions (see Fig. 3d). We modified several passages in the manuscript to make this aspect clearer. Additionally, we validate the model results for long-term plasmid evolution with our comparative genomics analysis where we ask: what do we see in contemporary plasmids, assuming that they can be considered the result of long-term plasmid evolution. Indeed, the comparative genomics analysis confirmed the presence of both partition and TA systems in most low-copy (conjugative) plasmids.

c. The discovery of the new TA serves to validate our suggestion for the presence of yet undiscovered TA systems in low-copy plasmids, with the example of IncX3 plasmids. We think that this result is important for our conclusions.

d. Finally, we think that our results and conclusions go beyond plasmid biology and evolution and are general for low copy extrachromosomal genetic elements – see our first discussion paragraph for a list of potentially relevant systems.

--O

2. Fig 4a and related analysis: a. Why did the authors analyze only plasmids from this limited taxonomic groups and did not include many other bacteria? b. I expected to see some statistical enrichment analysis for co-occurrence of TA and Par in the same plasmid. This was not shown for some reason.

Response: a. Plasmids in *Klebsiella*, *Escherichia* and *Salmonella* are the most extensively studied hence we expected that their sequences will be best annotated (e.g., with regards to partition and TA systems). We now added this reasoning in the manuscript.

b. considering the high abundance of plasmids harbouring both systems, conducting an enrichment test does not seem relevant to us. To make that clearer, we modified Fig. 4A where we replaced the subplots of Par and TA presence with an alternative plot showing the distribution of Par and TA presence/absence categories.

--O

Reviewer #2 (Remarks to the Author):

In this paper, the authors explore the strategies that low-copy conjugative plasmids use to persist within bacterial populations, focusing on the importance of active partitioning and toxin-antitoxin (TA) systems. By conducting experimental competitions between cleverly engineered plasmid variants and employing mathematical modeling, the authors demonstrate that the combination of partitioning and TA systems significantly enhances the stability and evolutionary success of low-copy plasmids. Genomic data analyses further confirm that these systems are very common in larger low-copy plasmids. Finally, the authors describe a new TA system that helps to explain IncX3 plasmid prevalence. The manuscript is clearly written, informative, and thoughtfully structured. The figures are clear, and the experiments and genomic analyses are generally sound.

Response: we are grateful for referee's kind words on our experimental system and for the helpful comments, all of which served to improve the manuscript.

--O

Our main concern is novelty. While we appreciate the authors' comprehensive approach, we suggest that the conclusions be more explicitly framed within existing knowledge. The role of TA/segregation systems in the persistence of large, low-copy-number plasmids is well documented (e.g., PMID: 12970556, 10931339, 26104442, 31187729, 26104348), and many observed trends, such as enrichment in larger plasmids, align with prior findings. The authors should clarify how this work advances beyond previous studies, particularly regarding the mechanistic insights from the experiments, to better highlight its contribution and originality.

Response: We thank the referees for this important comment. Indeed, others have noted that plasmids may harbour both active partition and TA systems (we cite several of those earlier studies). The main novelty in our work is the demonstration of the effect of those systems on plasmid fitness, thus supplying an evolutionary explanation for those observations. Accumulating evidence shows that the evolutionary success of plasmids in bacterial populations depends on drift and selection that occur in two organizational levels: the hosting cell and the population of hosts. The current study builds in our earlier conceptual demonstration of plasmid Darwinian individuality (Hülter et al. 2020; ref. 6), while investigating the effect of plasmid traits (Par and TA systems) on plasmid fitness. Our results are relevant for other extrachromosomal genetic

elements, beyond plasmids (examples are supplied in the first discussion paragraph). That being said, this comment led to several modifications of the manuscript, including the title, abstract, introduction and discussion.

We thank the referee for suggested references above. Indeed, some of those (but not all) indicate the presence of both partition and TA systems in plasmids, but none of those demonstrates the effect of those systems of plasmid fitness. Two of the references have been already cited in our manuscript. We generally prefer to cite more recent reviews on the systems, which is why we opted for a review from 2022 on TA systems and the Bouet and Funnel review from 2019 over the (excellent) Baxter and Funnel (2014). We opted to add Hernández-Arriaga et al. (2014) as we think that it includes several aspects not covered in the more recent review.

--o

Also, some results are not statistically supported. Although the differences depicted in the plots probably will be statistically significant, it would be beneficial to conduct the appropriate tests to confirm. Some (but not all) of these cases are identified below.

Response: we added statistical tests where we think is required (see below).

--o

We enjoyed the experimental system. It is thoughtful, comprehensive, and well-suited to the question. However, the model plasmid is small (~5Kb). This raises concerns about how the results might change if the authors used a large, low-copy plasmid. Perhaps this could be demonstrated using the model.

Response: this is a fair point and we clarify that in the manuscript, both the introduction and the discussion. Plasmid copy number is an important determinant of plasmid segregation probabilities (and plasmid allele dynamics), while no such evidence exists for plasmid size. Indeed, plasmid size and plasmid copy number is inversely associated, but this association is very difficult to disentangle. In mathematical models, plasmid size is (as far as we know) not considered as a relevant property that has an effect on plasmid segregation dynamics.

--o

The discussion (or intro) section could be enriched by citing/commenting on the ideas exposed by these papers (PMID: 15734695, 11058151), which expose a different view of TA systems.

Response: These studies are indeed highly relevant for the function of TA systems in plasmid evolution. We cited the earlier work by Cooper and Heinemann (2000) in our manuscript where we report the results of plasmid competitions between plasmids with or without TA, where our results are in agreement with those studies. We now cite also the later study (Copper and Heinemann 2005), although the results are similar. If the referee refers here to views on the role of TA in exclusion of competing plasmids (as in Cooper and Heinemann 2000) – we do not mention those as the plasmids in our setup are of the same backbone – hence we do not have any relevant new results towards that subject (and we prefer to keep it that way).

--o

Line 97 and thereafter. The terms homogenic and isogenic are used interchangeably. We think it is more common to use isogenic, but in any case, the authors should stick to one for clarity.

Response: the two terms are not the same. We used 'isogenic' to describe genetic identity, while 'homogenic' is used throughout to describe bacterial populations or cells harbouring a single

plasmid variant. In order to avoid misunderstandings, we now replaced two of the 'isogenic' used in the results section with 'identical'.

--0

Line 105. Maybe adding the plasmid notation here is useful: "The stable pCON genotype (pCON-S)"

Response: We do not have it there since it may be confusing to add the plasmid name before we explain the antibiotic resistance markers.

--0

Line 116. Statistical assessment of these results should be shown in Fig S1.

Response: That figure shows 2xSE which is comparable to 95% CI. We now added a statistical test of the results presented in Fig. S1 and amended the report in the results section accordingly.

--0

Line 117. According to Supp. Table S1, pCONn-PT has a PCN of approximately 2.4, not three. Is this value perhaps a bit low for a small plasmid?

Response: Indeed. Our PCN evaluation is based on sequencing results and this corresponds to a clear outlier. This is the reason why we use the average PCN in the model.

--0

Lines 119-126. Statistical analyses are lacking. I'm not an expert, but maybe this data could be analyzed using a Log-rank test or similar approach.

Response: we respectfully disagree, we do not compare the rate of plasmid loss, rather the plasmid frequency in the population. Plasmid loss is deterministic process and results in a differential decrease in the proportion of hosts for the different plasmid variants. We modified that statement to convey our conclusion clearer and avoid insinuation of a statistical conclusion. The simulation results parametrized with that data confirm the different trends of plasmid loss among the plasmid variants (Fig. 3c).

--0

Lines 131. A reference to Figure 1D is needed here.

Response: Done.

--0

Fig1c. Is there any reason why stability assays do not start at 1, i.e., 100% plasmid carriers?

Response: we are grateful for this comment. These frequencies were not corrected for plating efficiency. The deviation from 100% hosts in the first day is due to reduced plating efficiency using the differential droplet plating approach that we used for evaluating the CFU counts. To evaluate the plating efficiency using droplet plating we compared the results of this approach to replica plating. Populations of pCONn-S and pCONc-S were used to inoculate overnight cultures in liquid LB media containing 10 ng/μl Km (for pCONn-S) or 10 ng/μl Cm (for pCONc-S). Following the overnight incubation, the cultures were diluted and plated on selective and non-selective LB agar plates (as described in the manuscript methods). The plating was performed with droplet plating on rectangular plates (as used in the plasmid loss experiments) or manually on non-selective LB

agar plates. After overnight incubation at 37°C the resulting colonies were ‘replicated’ onto a selective agar plate using a velvet stamp. The difference between the replica plating and the droplet plating was on average 13.6% for pCONn-S and 6.1% for pCONc-S (see full dataset below), validating a reduced plating efficiency using the droplet approach. Consequently, the results of the plasmid loss experiments (Fig. 1C) were adjusted according to the plating efficiency as evaluated from the data of t_0 . This modification has no effect on the modelling parametrization and results.

Replica plating results			
Plasmid	Replicate	Delta	
pCONn-S	1	0.00E+00	
pCONn-S	2	0.00E+00	
pCONn-S	3	0.00E+00	
pCONc-S	1	0.00E+00	
pCONc-S	2	0.00E+00	
pCONc-S	3	0.00E+00	

* measured by superimposing the CFUs growing on selective and non-selective plates.

Differential droplet plating results								
Plasmid	Replicate	Plate_type	Count	CFU/mL	Difference	Average difference	Average %difference	
pCONn-S	1	LB	52	1.04E+09	-2.80E+08	-1.60E+08	-13.56	
pCONn-S	2	LB	83	1.66E+09	3.00E+08			
pCONn-S	3	LB	42	8.40E+08	-5.00E+08			
pCONn-S	1	Km	66	1.32E+09				
pCONn-S	2	Km	68	1.36E+09				
pCONn-S	3	Km	67	1.34E+09				
pCONc-S	1	LB	80	1.60E+09	-3.00E+08	-1.00E+08	-6.07	
pCONc-S	2	LB	96	1.92E+09	1.80E+08			
pCONc-S	3	LB	71	1.42E+09	-1.80E+08			
pCONc-S	1	Cm	95	1.90E+09				
pCONc-S	2	Cm	87	1.74E+09				
pCONc-S	3	Cm	80	1.60E+09				

--0

Fig1d. None of the plots show pCONn-S/pCONc-U, it’s something mislabeled here?

Response: Indeed! Many thanks for noting this error. It has been corrected in the revised version.

--0

Lines 159-176 and accompanying Supp. Figures. The statistical assessment of these results in provided in Fig 2b. which should be referenced before and explicitly.

Response: Done. The legend of Figure 2 includes such an explicit reference as well.

--0

The authors should compete pCON-PT against pCON-P and pCON-T. This would help to show whether the combination of both systems is better than each separately.

Response: such competitions are impossible to perform as the identical sequences of the Par and TA loci would result in frequent recombination of the competing plasmids. This is the reason why we designed the plasmid competitions such that plasmids with stability traits are competed against the unstable and stable plasmid variants, and the results of those competitions were used to infer the expected results in competitions of the PT vs. P or T variants – which is what is presented in Figure 2b.

--0

Figures are referenced inconsistently in Lines 179-194. Should it be Fig2.*A*6?

Response: This was indeed a typo. Corrected.

--0

L186. This sentence needs rewriting.

Response: We rephrased that sentence. Thank you.

--0

L222. I'm not an expert on mathematical models, but the lines about PCN are confusing to me. I don't understand why N_c is defined as $2/3 N_{pcn}$, and N_{pcn} is defined as $3/2 N_c$. Can you elaborate on this further? Additionally, Fig 3b does not seem to be related to PCN. Perhaps it would be a good idea to include an additional row showing PCN for all plasmids.

Response: We thank the referee for this comment. We now simplified the model by using the mean measured PCN as the copy number parameter. The previous assumption of the model estimated the mean copy number over the cell cycle.

--0

L297: There is a typo; a comma is used where a point should be.

Response: Corrected.

--0

L345: There is a typo in "Vibrio cholera".

Response: Corrected. Thank you!

--0

L411. I do not understand how this conclusion is supported by the results presented in this paper. Please elaborate or eliminate.

Response: Thank you for this comment. To clarify our conclusion on the state of plasmid heteroplasmy, we rewrote and elaborated that paragraph in the discussion.

--0

Reviewer #3 (Remarks to the Author):

Effe et al. present a set of experimental and theoretical analyses of bacterial plasmids with varying persistence strategies. The authors constructed variants of a small plasmid with different levels of stability, by introducing two alternative persistence mechanisms, involved either in plasmid

partitioning to daughter cells (parA), toxin-antitoxin production killing plasmid-free daughter cells (parB), or both. They then perform pairwise “intracellular competitions” among these plasmids and a stable and unstable control variant, by measuring the segregation of both types from heteroplasmic cells after 5 days, showing the superiority of plasmids with both persistence mechanisms over single mechanisms. These results are compared with predictions from a mathematical model, which provide qualitative support for the experimental findings. Finally, a survey of thousands of plasmid genomes in a public data base shows that plasmids larger than 19 kbp often encode both persistence mechanisms. The authors conclude from these combined analyses that the combination of toxin-antitoxin and partitioning mechanisms is most effective for the persistence of low-copy plasmids.

I think the experiments with constructed plasmids is quite elegant, and the combination with modelling and analyses of a large set of available plasmids makes their conclusion solid and convincing. However, I cannot judge how surprising and novel this conclusion is. Also, I do have a few questions and comments, which I hope may help to clarify and perhaps solidify their findings and conclusion.

Response: we thank the referee for the accurate summary of our study and for the kind words. The comments of the referee were very helpful and led to several modifications and additions to the manuscript. Furthermore, we now adopt the (Nowack 1987) terminology used here – homoplasmic and heteroplasmic – instead of homogenic and heterogenic.

--O

1. The authors say they perform intracellular competitions with these plasmids. However, their model suggests that competition among homoplasmic bacteria makes up most of the competition (e.g. Fig. 3D, where heteroplasmic cells disappear within ~10 generations). Therefore, differences in host fitness effects have the potential to strongly impact the outcome of competition, but I didn't see how this was taken into account. It is only said (lines 114...) that “Most plasmids have a negligible effect on their host fitness..”, but in the next line a rather substantial fitness cost of up to 5.5% is mentioned for 2 of the plasmids. However, in the model fitness costs did not seem to play a role. Have the authors considered how the measured fitness costs of the plasmids affected their results?

Response: We thank the reviewer for this important comment. Indeed, two plasmids (pCONn-T and pCONc-U) exhibited modest but statistically significant fitness cost of ca. 5%. However, the remaining plasmids, including the respective reciprocal plasmids pCONc-T and pCONn-U, showed no measurable effect on host fitness (see Fig. S1, where we now added also a statistical test). Based on this, we concluded that overall, the fitness effect of most model plasmids on their host is minor for most plasmids in our study.

With respect to the specific concern raised: competition between the two unstable plasmids pCONn-U and pCONc-U show no significant difference in their fitness ($P=0.115$, using paired sign test; moved from Fig. S3 to Fig. S2A in the revised manuscript), hence we conclude that the negative fitness effect of pCONc-U on their host had a small effect on the plasmid competition outcome. Plasmid competitions against pCONn-T yielded the same or similar outcome in the reciprocal competitions, hence we think that also here the negative fitness effect of that plasmid has little effect on the plasmid competitions. Taken together, the observed (small) differences in plasmid effect on the host fitness do not qualitatively affect the competitive hierarchy among plasmids (as presented in Fig. 2b). We now supply these detailed explanations in the results section.

As for the model, we deliberately chose not to include plasmid-specific host fitness effects for two reasons: (1) to keep the model as simple and general as possible by assuming all

resistance markers to have the same host fitness effect; and (2) because comparing the fitness of heteroplasmic and homoplasmic hosts is experimentally very challenging. Since heteroplasmic cells disappear rapidly (within ~10 generations, Fig. 3d), and the observed fitness hierarchy aligns with plasmid-level dynamics rather than host-level effects, we consider this simplification to be justified. We have now clarified this rationale in the revised manuscript to address the reviewer's concern directly (2nd paragraph in the model results).

--O

2. In line 169, it is mentioned that heteromultimers of competing plasmids were sometime observed at high frequency. However, it was unclear how this was taken into account, while this may substantially affect the outcome of competition. Also, it may be informative to add this info to Fig. 2.

Response: Thank you for this comment. Indeed, one reason of the pCON-U instability is its tendency to form multimers (see Wein et al. 2019; ref. 1 in the response below). Furthermore, previously we observed that model plasmids of the pCON backbone tend to form heteromultimers, and those multimers can be stable over time (Hülter et al. 2020; ref. 2 in the response below). Therefore, we suspected that multimer formation is a property of the pCON backbone rather than a frequent event in plasmid evolution. Consequently, we performed a large-scale study of plasmid duplication in *Escherichia* (using the same KES dataset used here; reported in Hanke and Dagan 2025 *bioRxiv*: [10.1101/2025.03.14.643293](https://doi.org/10.1101/2025.03.14.643293), in revision). Since we observed plasmid multimers only rarely in *Escherichia* strains, we consider the presence of pCON (hetero)multimers a specific property of the pCON backbone in our experimental setup. We now added this information in the methods section.

To decrease the effect of such multimers on the results of plasmid competitions, we performed the plasmid competitions in a *recA* knockout mutant. Since the heteromultimers have a stable presence in that host background (as we show in Supp Fig. S4), we think that they do not have a large effect on the outcome of plasmid competitions that occur early on, directly after the introduction of the competing plasmids into the host. In our setup we compensate for their emergence by multiple replications of the competition experiments and the reciprocal competition design. We do not think that it is informative to add that information in Fig. 2.

--O

3. More broadly, did the authors sequence some of the end products of competition to see if other “mutations” may have affected their results? For example, could plasmid integration in the chromosome or general fitness enhancing mutations have affected their results?

Response: Indeed, we sequenced several of the end populations in the evolution experiment. Two examples of such sequencing results are reported in the manuscript in Supplementary Figure S4. We never observed mutations in the model plasmids (beyond multimers and heteromultimers). Furthermore, we never observed pCON integration into the chromosome backbone, neither in this study nor in our previous experimental evolution experiments with that plasmid backbone (refs. 1-4 below). We would like to note that plasmid integration in the chromosome is generally very rare (ref. 5 below).

1. Wein T, Hülter NF, Mizrahi I, Dagan T. Emergence of plasmid stability under non-selective conditions maintains antibiotic resistance. *Nat Commun*. 2019;10: 2595–13. doi:[10.1038/s41467-019-10600-7](https://doi.org/10.1038/s41467-019-10600-7)
2. Hülter NF, Wein T, Effe J, Garoña A, Dagan T. Intracellular competitions reveal determinants of plasmid evolutionary success. *Front Microbiol*. 2020;11: 2062. doi:[10.3389/fmicb.2020.02062](https://doi.org/10.3389/fmicb.2020.02062)

3. Wein T, Wang Y, Hülter NF, Hammerschmidt K, Dagan T. Antibiotics interfere with the evolution of plasmid stability. *Curr Biol.* 2020;30: 3841-3847.e4.
doi:[10.1016/j.cub.2020.07.019](https://doi.org/10.1016/j.cub.2020.07.019)
4. Wein T, Wang Y, Barz M, Stücker FT, Hammerschmidt K, Dagan T. Essential gene acquisition destabilizes plasmid inheritance. *PLOS Genetics.* 2021;17: e1009656.
doi:[10.1371/journal.pgen.1009656](https://doi.org/10.1371/journal.pgen.1009656)
5. Kadibalban AS, Landan G, Dagan T. The extent and characteristics of DNA transfer between plasmids and chromosomes. *Curr Biol.* 2024;34: 3189-3200.e5.
doi:[10.1016/j.cub.2024.06.030](https://doi.org/10.1016/j.cub.2024.06.030)

--o

4. What was not clear to me from the survey of sequenced plasmids (Fig. 4), is why both persistence mechanisms were typically found together on larger plasmids that often were conjugative. First, conjugation is another mechanism enhancing plasmid persistence, so these plasmids would be in less “need” for these other persistence mechanisms. Second, was there a correlation with plasmid size, such that the likelihood of having both mechanisms on the same plasmid would be larger than expected from randomly distributing mechanisms across plasmid DNA? One may also compare the frequency of combinations of specific mechanisms of both types (portioning versus toxin/antitoxin) with their expected frequency if they had no combined benefit by multiplying their individual frequencies across plasmids.

Response: We thank the referee for raising this important aspect! First: yes, our results mean that conjugation is not sufficient for plasmid persistence. This is because vertical inheritance typically dominates over plasmid transfer in plasmid evolution. We added few sentences on that in the last discussion paragraph. Second: the question regarding size is a tricky one to answer, because conjugative plasmids are generally larger due to the conjugation machinery, and also because in order to account for plasmid lineages (plasmid taxonomic units; PTUs) one would have to perform such an analysis within PTUs, but plasmids in the same PTUs typically have similar sizes. In other words – at the moment we cannot answer this question with confidence. Lastly: testing for statistical patterns of different Par and TA combinations is definitely an interesting research direction. We have done that using the KES dataset during the work on this manuscript but so far we did not find patterns that are worth reporting (. It may require a larger dataset.

--o

5. It may be good to clarify where the experimental system is described, that the horizontal transfer of these plasmids does not play a role. Also the size of plasmids may be given.

Response: Right. We added this information in the second results paragraph.

--o

6. Line 182-183: Fig. 2.6 and 2.7 should be Fig. 2A.6 and 2A.7.

Response: Corrected. Thank you.

--o

7. In the description of model parameters, the parameter for plasmid copy number is first said to be n_r (line 202), but further down (line 208) n_c is used for plasmid copy number?

Response: Thank you for this comment. In our previous MS version, we used n_c for the number of copies before cell division while n_r is the copy number parameter of our math model. Now we

use only copy number n_c while the copy number before cell division is $2n_c$ in both cases mentioned.

--0

8. Line 260-262: “.. the stable plasmid variant (S) has a higher persistence in the very long term compared to the PT-plasmid”. Where is this shown? Fig. 3 suggests differently.

Response: this was indeed not shown. In the revised manuscript we changed the copy number as we simplified the model by taking the mean plasmid copy number as copy number parameter, $n_c = \text{Round}(\text{PCN}) = 5$ (previously $n_c = \text{Round}(2/3 \text{ PCN}) = 4$). Using $n_c = 5$ yields more stability for the S variant and the current Fig 3B shows that S is indeed more stable than PT in the long-term.

--0

9. At several places, grammar is incorrect. For example the heading in line 266 is not understandable, similar in line 299, 347, 354.

Response: We modified those sentences. Thank you for this comment.

--0

Response to referees' comments on Effe et al. 2025.

Reviewer #2 (Remarks to the Author)

Overall, we're pleased with the authors' response. The manuscript is now clearer and better framed within the existing literature. The authors have effectively addressed concerns regarding novelty and statistical issues.

However, some issues still require attention. The authors state that direct comparisons between the PT plasmid and the P or T plasmids could not be conducted due to the risk of recombination at identical loci. Nevertheless, all pCON variants share the same backbone. If recombination is truly frequent enough to preclude PT vs. P or PT vs. T competitions, it raises the question of how it was ruled out as a confounding factor in the other experiments. A brief mention of the potential effect of recombination on the observed results should be sufficient to address this issue.

Response: this is a slight misunderstanding. The two pCON-U backbones are not exactly identical as they comprise the two different antibiotic resistance markers – *nptII* and *cat* – see illustration in Fig. 1a. Our conclusion of the PT superiority is based on the result of the comparisons of P, T, and PT variants to both U and S variants – as shown in figure 2b. We think that this design of our experiments does not require further explanations on the possibility of recombination.

--0

Furthermore, the abstract indicates that the combination of partitioning and TA systems is more effective than either strategy alone, implying that this comparison was directly tested. In reality, the advantage of the combined PT system was inferred from modeling and indirect competition results, rather than being experimentally demonstrated through direct PT vs. P or PT vs. T competitions. While the inference is well supported, the abstract could be phrased more cautiously to acknowledge this limitation.

Response: we respectfully disagree with this comment. Our formulation in the abstract is rather careful ('we find') in order not to insinuate a direct observation ('we show / demonstrate'). The combination of our experimental results, the modelling, and the comparative genomics support our conclusions.

--0

We anticipate these issues to be easy to solve, and we congratulate the authors on a strong and well-executed study.

Response: we thank the referees for their useful insights and suggestions.

--0

Reviewer #3 (Remarks to the Author)

I thank the authors for their responses to my comments. All responses were reasonable and take away most of my concerns. With respect to my comment 4 about large (often conjugative) plasmids typically having both plasmid persistence mechanisms (TA and partitioning), I can see that it may be too simple to perform the statistical analysis I suggested given the effect of conjugation genes on plasmid size. However, it may still be informative to the reader to mention -- and possibly briefly discuss -- this association with plasmid size.

Response: We are happy that the referee found our response reasonable. To add information on the association between plasmid size and mobility we now added a small section in the discussion (one before the last paragraph) that reads as following: Large plasmid size is

typically associated with self-transmissibility or mobility where the presence of conjugation and mobility modules contributes to the increase in plasmid size¹. Indeed, not all large plasmids in our analysis are conjugative (Fig. 4a). Nonetheless, recent studies suggest that large plasmids in several enterobacteriaceae PTUs annotated as mobilizable or non-mobile evolved from conjugative plasmid ancestors². For example, PTU E5 that comprises mostly non-mobile plasmids was inferred to be related with PTU FE that comprises mostly conjugative plasmids. Plasmids in PTU E5 comprise multiple pseudogenes, including non-functional copies of conjugation genes, and are found in a narrow host range, both are characteristics of domesticated plasmids³. The prevalence of non-functional transfer (*tra*) gene copies in other PTUs of large plasmids in *Escherichia* and *Salmonella*³ suggests a common evolutionary origin, with PTUs of large mobilizable or non-mobilizable plasmids having been derived from conjugative plasmid ancestors.

Additionally, we modified the abstract to make it more clear that our conclusions pertain to large plasmids including conjugative and mobilizable types.

--O

1. Smillie, C., Garcillan-Barcia, M. P., Francia, M. V., Rocha, E. P. C. & de la Cruz, F. Mobility of plasmids. *Microbiol Mol Biol Rev* **74**, 434–452 (2010).
2. Coluzzi, C., Garcillán-Barcia, M. P., de la Cruz, F. & Rocha, E. P. C. Evolution of plasmid mobility: origin and fate of conjugative and nonconjugative plasmids. *Mol Biol Evol* **39**, msac115 (2022).
3. Hanke, D. M., Wang, Y. & Dagan, T. Pseudogenes in plasmid genomes reveal past transitions in plasmid mobility. *Nucleic Acids Res* **52**, 7049–7062 (2024).

Reviewer #4 (Remarks to the Author)

Response: we thank the referees for their helpful comments.

--O